# Soil Organic Matter in Urban Areas of the Russian Arctic: A Review

**Evgeny Abakumov** [1,*] **, Alexey Petrov** [2] **, Vyacheslav Polyakov** [1] **and Timur Nizamutdinov** [1]

1   Department of Applied Ecology, Faculty of Biology, St. Petersburg State University, 16th Liniya V.O., 29, St. Petersburg 199178, Russia; slavon6985@gmail.com (V.P.); timur_nizam@mail.ru (T.N.)
2   Laboratory of Permafrost Soils North-Eastern Federal University in Yakutsk, Belinsky Str., 58, Yakutsk 677027, Russia; petrov_alexey@mail.ru
*   Correspondence: e.abakumov@spbu.ru or abakumov@mail.ru

**Abstract:** Polar ecosystems are the most important storage and source of climatically active gases. Currently, natural biogeochemical processes of organic matter circulation in the soil-atmosphere system are disturbed in urban ecosystems of the cryolithozone. Urbanized ecosystems in the Arctic are extremely under-investigated in terms of their functions in regulating the cycle of climatically active gases. The role of urban soils and soil-like bodies in the sequestration and stabilization of organic matter is of particular interest. The percentage of gravimetric concentrations of organic matter in Arctic urban soils are almost always determined by the method of dichromate oxidation and are subject to extreme variability (from tenths of a percent to more than 90% in man-made soil formations), but the average carbon content in the surface soil horizons can be estimated at 5–7%. The surface humus-accumulative horizons are represented by a variety of morphological forms with the content of organic matter of various origins. The work also focuses on those forms of organic matter, the content of which is extremely small, but very important for the biogeochemical functioning of soils-polycyclic aromatic hydrocarbons and components of petroleum products, as well as labile forms of soil organic matter. We recommend that further studies of the organic matter system be conducted in urbanized areas since the carbon cycle there is severely disrupted, as well as carbon flows. The urbanization and industrialization processes in the Arctic are progressing, which could lead to a radical transformation of carbon ecosystem services.

**Keywords:** urbanization; urban soils; permafrost-affected soils; carbon sequestration; ecosystem services; Arctic

## 1. Introduction

According to the annual report of the World Meteorological Organization [1], the climate continues to change everywhere from the mountaintops to the ocean depths. In 2022, droughts, floods, and heat waves affected populations on all continents and resulted in multibillion-dollar losses. Antarctic sea ice levels have fallen to record lows, and the melting of some European glaciers has reached literally unprecedented rates. Permafrost thawing occurs globally in the mountains and on the plains of the Arctic, resulting in the release of huge amounts of climatically active gases into the atmosphere and biogenic elements into the hydrosphere [1]. Permafrost soils play an enormous role in the accumulation and deposition of organic carbon in the Earth's pedosphere, and permafrost soils are crucial to future climate change.

Although permafrost soils store carbon, the amount is variable, the average stock in a 1 m soil section is evaluated as 29 kg m$^2$ in soils of the Arctic and 8 kg/m$^2$ in Tibet's landscape [2]. The stocks of soil carbon in Antarctica vary from 0 to 35 kg/m$^2$ [3], where zero is typical for ahumic soils (regoliths), and the highest percentages are for active and abandoned penguin rookeries. The total stock of soil organic carbon within 0–3 m in Arctic soil is estimated as 1014 Pg C [2]. This is a carbon stock, but not soil organic matter and

scientists are faced with the problem of calculating soil organic matter with the use of the coefficient–Corg $_X$ K = SOM, where Corg is the carbon of soil organic compounds [4], K is the empirical coefficient, and SOM is soil organic matter. The empirical coefficient is determined by the nature and elemental composition of organic molecules; this coefficient was proposed by E. Wolf in 1864 based on the information available at the time about the carbon content of humic acid, equal to 58% on average, and is currently used. [5–7]. For the Arctic, especially for its Siberian part, there are almost no data to calculate this coefficient. The carbon content of humus in different types of soils is different and varies widely: 36–62%. This leads to even greater uncertainty in the estimates of organic matter stocks, and it therefore reduces the accuracy of model estimates of the global carbon budget and does not contribute to the adequate assessment and monetization of carbon ecosystem services [8]. Currently, estimates of the sequestration and storage of organic carbon in the permafrost soil layer remain not quite accurate and are characterized by significant errors [9]. This is especially important if one takes into account the fact that the carbon of permafrost-affected soil plays a crucial role in the feedback of climatic dynamics on permafrost carbon changes [10].

Currently, intensive industrialization and urbanization are taking place in the Arctic, which leads to the degradation of permafrost, the formation of new types of soil and vegetation, and changes in the rate of mineralization of organic matter and the degree of its humification. In this regard, it is important to analyze the extent to which soils of urbanized ecosystems differ from background ecosystems in terms of carbon content and carbon stores.

In the process of urbanization, the plant community changes significantly in Arctic cities. Currently, the plant community's changes include the borealization of Arctic flora [11]; the invasion of more southern, including invasive, plant species [12]; and the increase in the proportion of woody and herbaceous species [13], which may lead to changes in the composition of humification precursors. Thus, it can be assumed that the composition of humus in urban soils will be different from that in background zonal soils.

Another poorly studied source of carbon in Arctic urban soils is black carbon, which is formed as a result of wildfires, human activities [14], and polycyclic aromatic hydrocarbon (PAH) [15]. We can only state that this process takes place, but we cannot estimate it quantitatively. Urbanization in the Arctic has long been accompanied by the development of tundra and forest-tundra soils and their involvement in agriculture, and this process has been more pronounced in Eurasia than in North America [16]. Thus, agricultural soils can be considered as a contribution of organic carbon to the soils of urbanized environments. Urban soils of polar environments can serve as a model for the rate of soil organic matter mineralization in the cryobiosphere with increasing topsoil drainability and changes in its albedo and vegetation cover characteristics.

Thus, the issue of the ways that organic carbon accumulates in the soils of urbanized ecosystems of the Arctic is an interesting and practically unexplored question, so the purpose of this review article was to analyze the primary data on the levels of organic carbon content and accumulation in the soils of Russian Arctic cities and the mechanisms of stabilization of these compounds.

## 2. Materials and Methods

The methodology consisted of work in the literature on the characterization of urban and background soils and adjacent ecosystems in the Russian Arctic, as well as our own field and laboratory studies (Figure 1). Our own field data were collected between 2013 and 2022 during various expeditions to the Russian Arctic. Usually, expeditions to the Arctic do not stay long in the cities, the scientist almost immediately goes to the field. In our case, the study of urban soils was given special attention. Thus, this review is a pioneering study for the Russian Arctic. It should be noted that the Arctic in Russia is highly urbanized and up to 85% of the population of the Russian Arctic belt is concentrated in cities and urban-type

settlements. North America, by contrast, is dominated by the "fly-in-fly-out" method of environmental development, which has not led to urbanization and an impact on nature.

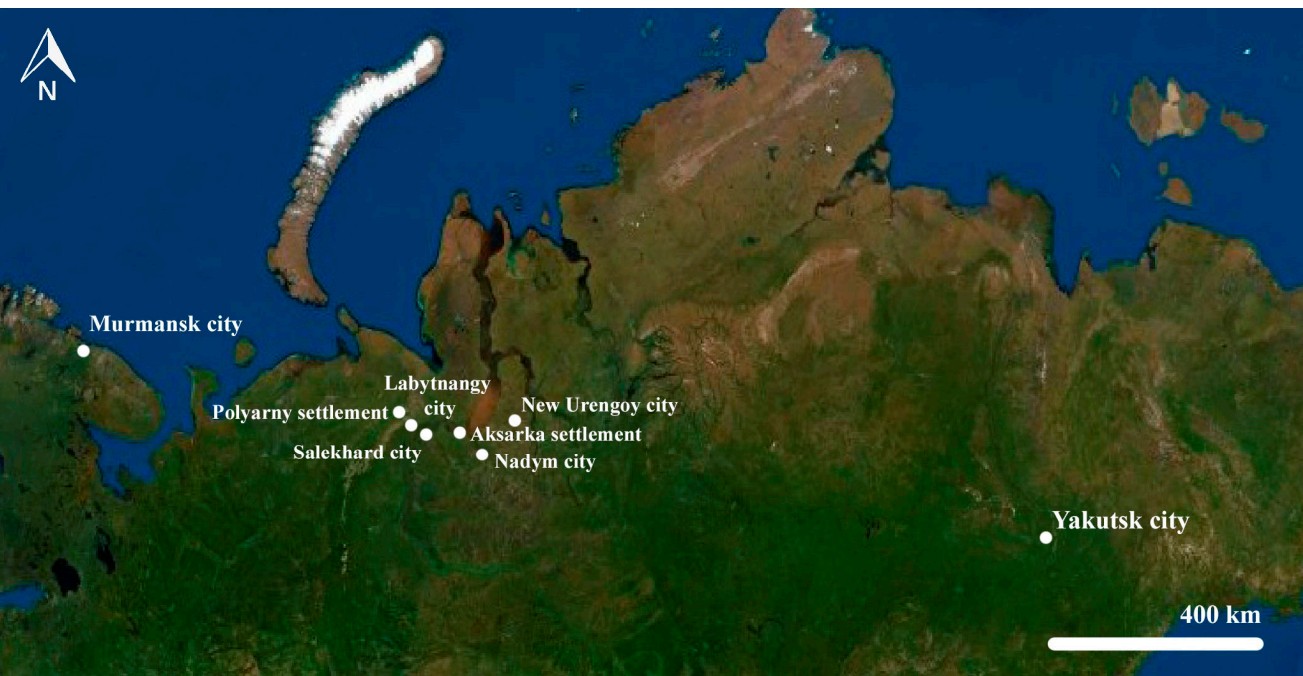

**Figure 1.** The area of studied cities in the Russian Arctic.

In the Department of Applied Ecology, we conduct basic chemical and physical-chemical analyses and study the micromorphological structure of soil. The chemical soil parameters were analyzed on a fine earth of soil after being passed through a 2 mm sieve. The chemical analyses were performed using classical methods: C and N contents were determined using an element analyzer (EA3028-HT EuroVector, Pavia PV, Italy) and pH in water and in salt suspension (soil-dissolvent ratios 1:2.5 in the case of mineral horizons and 1:25 in the case of organo-mineral horizons) using a pH meter (pH-150M Teplopribor, Moscow, Russia). The particle-size distribution of soils was determined by the Kachinsky method, a Russian analog of the analysis proposed by Bowman and Hutka [17]. For micromorphology, fine sections of soil material were prepared from micro monoliths of soils sampled in the field. Samples were dried and saturated with resin. The obtained thin sections were analyzed using a polarizing microscope (Leica DM750P, Mannheim, Germany) with LAS 4.9 software in plane-polarized light and crossed polarized nicols.

### 3. Results and Discussion

*3.1. Morphology of Organic Matter*

The top layer of zonal soils of the Arctic is usually represented by the folic horizon [17,18], which can be classified into different levels of soil layers ($O_1$, $O_2$, $O_3$) in accordance with the Russian soil taxonomy [19]. Overwatering leads to the formation of Histic layers with increased thickness and a lower degree of decomposition of organic matter. Only in some drained soils of the southern tundra or on the slope of the southern part of the Polar Urals, an umbric horizon, designated as AY in the Russian soil taxonomy, can be found in the surface part of the soil section.

The first distinctive feature of urban ecosystems is the destruction of the upper organogenic soil horizon; it is replaced by umbric horizons of varying thicknesses. Most often, it is 5–10 cm, but in the soils of the cities of Nadym and Novy Urengoi in Western Siberia, in the park zones, the thickness of the humus horizon can reach 20 cm or more. This is associated with the input of organic matter, mainly of peat origin, during landscaping, construction of the park soil layer, and cold temperature. For urban soils of the

city of Murmansk, folic horizons have been identified [20]. In general, urbanization leads to the diversification of surface soil horizons, with the underdeveloped humus horizon or Folic layer being replaced by darker plaggic, urbic, or transportic humus material of different origins (Figure 2). Urbic layers are characteristic of the archaeological settlement of Labytnangi [21] and the parks of Novy Urengoi [22]. The plaggic soil horizon is not rare for northern soils and is documented for subarctic regions [23]. Urbic soil horizons with variable organic matter content are typical for old European cities [24], but even in young Arctic settlements, they exist in cities 40–50 years old. Transportic soil horizons may develop as a result of reclamation practices [25]. Urbanized cryogenic ecosystems are characterized by the practice of olericulture in small areas of soil in and around cities [22], which led to the formation of plaggic soil horizons. In the gardens of the Vorkuta city [25], an accumulation of humus and darkening of the upper urbic layer of soil were noted.

Plantation plowing leads to a significant change in the morphology of the soil profile, which was a regular phenomenon in the Subarctic during the Soviet period. This horizon can be designated as turbic (TUR), and soils are identified as cryoturbozem (Turbic Cryosol in WRB) in the Russian soil taxonomy [26]. It can be highlighted that the thickness of humus-enriched horizons of urban soils in Yamal and in the European part of the Arctic [27] is higher than in Yakutsk [28]. Thus, soil formation in Arctic cities is expressed not only in cryoturbation or stagnation, which is typical of natural tundras, but also in the formation of humus-enriched horizons, which leads to the formation of darker-color humus horizons being generally characterized by a lighter particle size distribution.

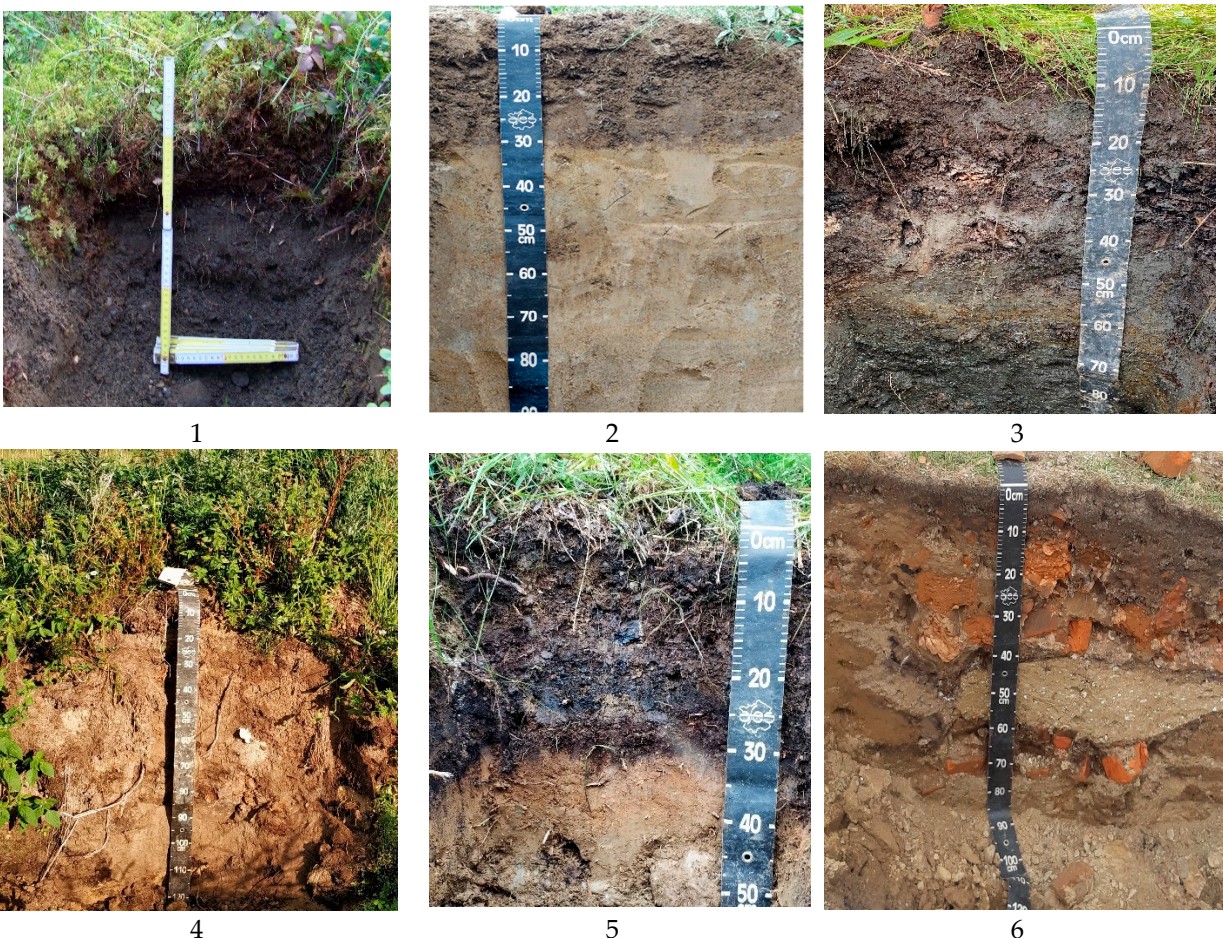

**Figure 2.** *Cont.*

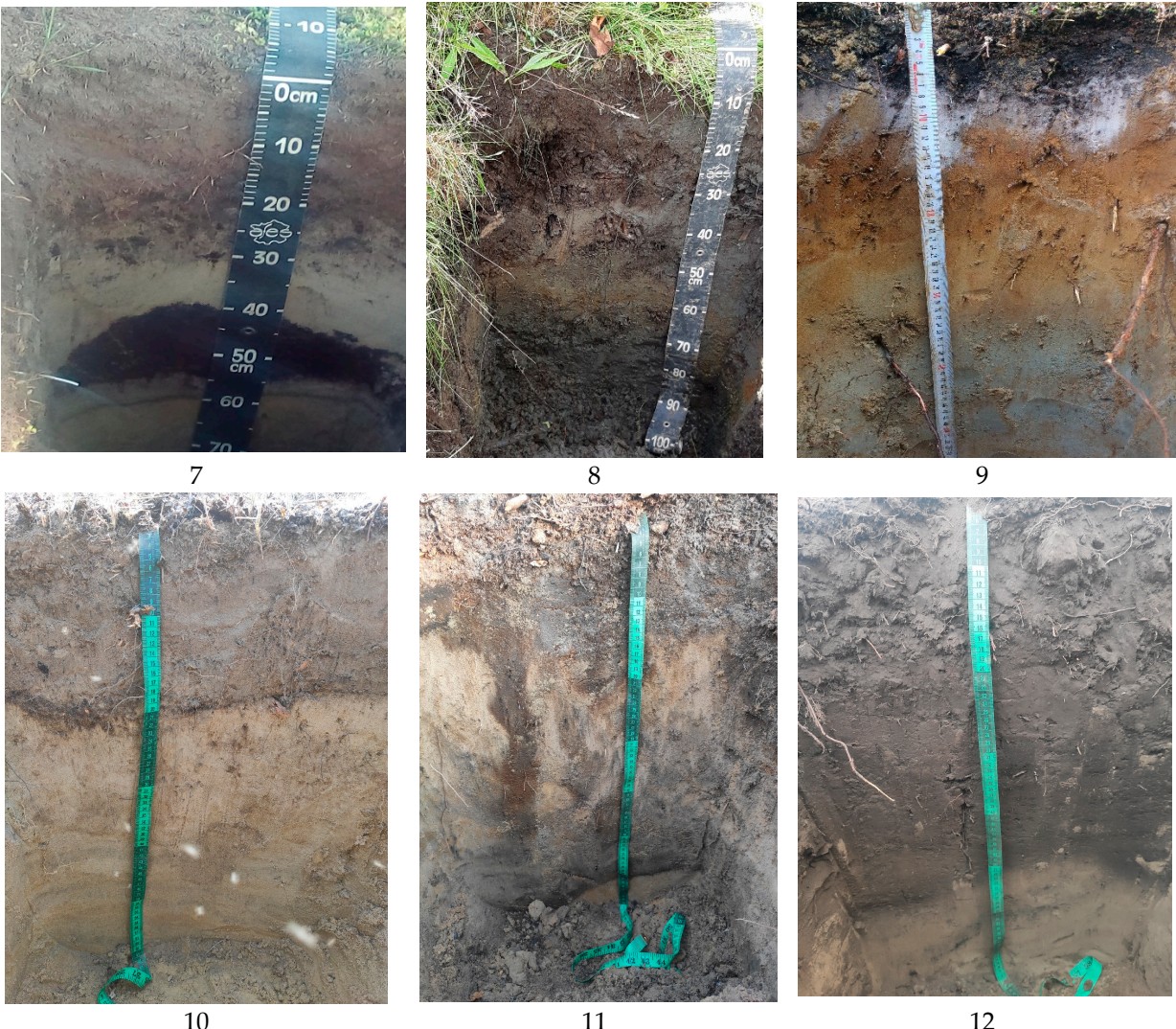

**Figure 2.** The macromorphology of topsoil humus horizons. Image 1—folic horizon of benchmark Cryosols; 2—vicinity of Salekhard city, plaggic horizon of the Anthrosols, Yamal research field station, Salekhard; 3—gray humus horizon of vegetable garden, Salekhard; 4—humus-enriched material for construction of Anthrosols in vegetation nursey, Salekhard; 5—humus horizon with pyrogenic features, vegetable garden in abandoned settlement Polyarny, Yamal region; 6—humus-enriched urbic horizon with archeological artifacts, Labytnangy city, Yamal; 7—deep plowing in former potato farm in vicinity of Nadym city, Yamal region; 8—plaggic soil horizon in abandoned fishery village of Tovogopol, vicinity of Aksarka settlement, Yamal region; 9—gray humus horizon of the park soil in New Urengoy city park, Yamal Region; 10—gray humus horizon of the fallow land in Yakutsk city; 11—pyrogenic soil of Yakutsk city; 12—experimental arable field of agriculture school near Yakutsk city.

One of the ways to study organic carbon is a micromorphological investigation of a soil thin section; here, we can observe the organic material in different forms, from non-humified organic tissues to highly decomposed organic matter. Only few data have been reported on the micromorphological characteristics of the soils of urbanized territories in the Arctic [29]. In thin sections of Yakutsk soils, an accumulation of pyrogenic carbon particles in the upper layers of urban soils has been noted. Organic matter in the soil is represented in various forms, both as undecomposed organic residues, which are more accessible to biodegradation, and as less-accessible organo-mineral complexes (Figure 3).

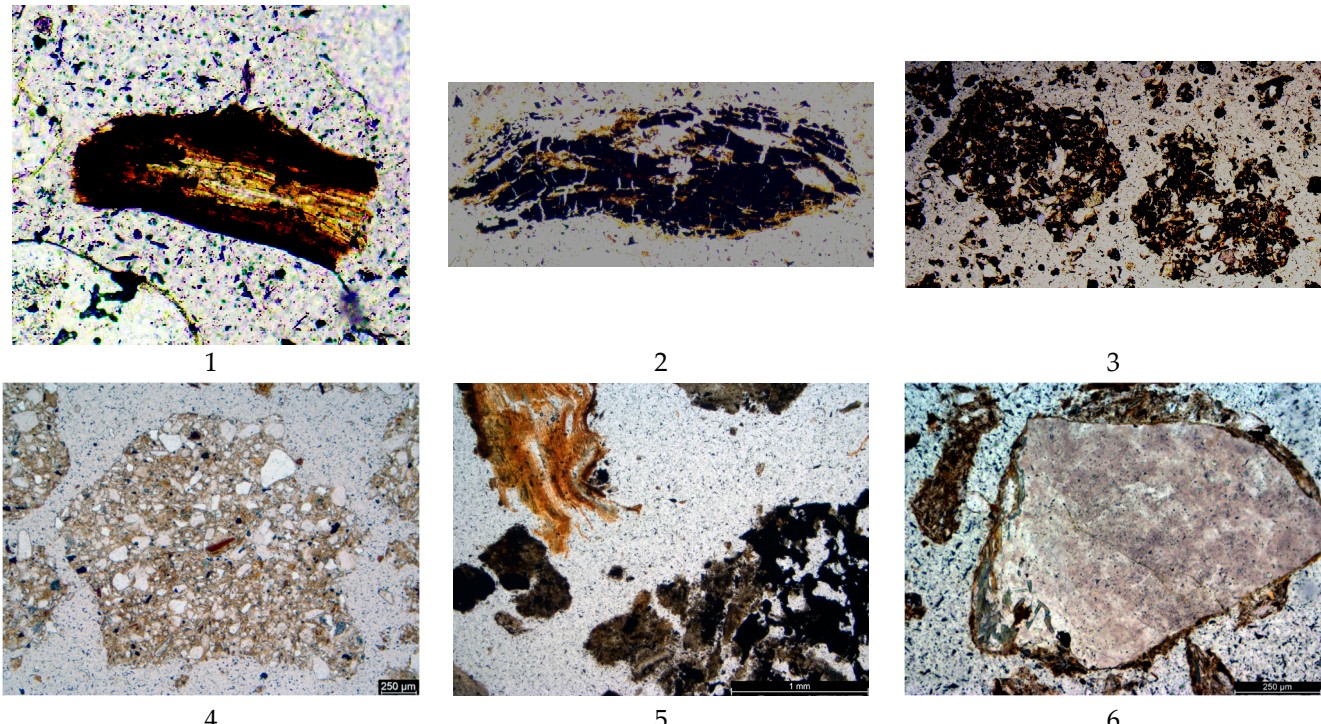

**Figure 3.** Soil micromorphology of anthropogenically transformed soils. Image 1—plant tissues in fallow land in Yakutsk; 2—pyrogenic carbon in soil of Yakutsk; 3—organo-mineral complex in experimental arable field of agriculture school near Yakutsk; 4—organo-mineral aggregate from the topsoil of the Yamal experimental agricultural station (Salekhard); 5—plant tissues of a weak degree of decomposition and a porous piece of charcoal filled with organic material (topsoil, Polarny settlement); 6—fractured mineral fragment superficially enveloped in organic and clay material (plaggic soil horizon, Tovopogol settlement).

Carbon in urban soils can be represented in the form of plant residues with various degrees of decomposition (Figure 2(1,5)), as well as different variants of pyrogenic genesis (Figure 2(2,5). In agricultural soils, especially with a long history of cultivation, organic carbon that was previously inherited into the soil with organic and organo-mineral fertilizers aggregates with the mineral soil matrix by forming organo-mineral complexes (Figure 2(3,4)). Normally, urban soils in the cryolithozone are subject to deep transformation and contain allochthonous carbon in their profile, in the form of particles of coal, plant tissues, and anthropogenic artifacts. As for organo-mineral interactions, they are weakly pronounced in the soils and are expressed in the accumulation of poorly decomposed organic matter, which indicates recent pedogenesis processes; i.e., there was not enough time for the development of organo-mineral interactions. Redoximorphic micromorphological features [30] and redistribution of organic matter within the soil profile due to cryogenic mass transfer [31], expressed in some cases in the accumulation of the second layer of organic matter at the boundary with the permafrost table, are observed in the natural tundra.

### 3.2. Gravimetric Concentrations and Profile Distribution of Organic Matter

The content of organic matter in urban soils varies significantly in the cities of the Russian Arctic. Thus, a high content of organic carbon was noted for the soils of Murmansk [20]. Thus, the carbon content was up to 10% in arable soils with a C/N ratio of up to 30–50, which agrees well with Petrova's data obtained for the soils of the city of Murmansk [27]. Even at a depth of about 80 cm, the organic carbon content was about 1%. The bulk density was 0.16 g cm$^3$ for the folic horizon and 0.90 g/cm$^3$ for the AU horizon of urban soils [20]. The city of Vorkuta [25] shows huge gravimetric percentages of soil carbon, especially in

the upper soil layers, in some cases up to 20%, which is due to the formation of constructed soils with the use of transported materials. The soil of the Vorkuta city is characterized by alkaline pH values, while in Murmansk soils in different areas were both acidic and alkaline. This should be noted in assessing the degree of stabilization of soil organic matter. It is notable that an important part of the organic matter of urban soils is benz(a)pyrene; its concentration reaches 1200 ng×g in the soils of Vorkuta [25]. This is not a high value in terms of volumetric carbon content, but this polycyclic aromatic compound is very biochemically active and is known to be carcinogenic, so it is important not only to estimate volumetric carbon stock but also its composition, which plays a very significant ecological role [32]. Another city in the Komi Republic with registered carbon content is Ukhta, located to the southwest of Vorkuta. The carbon content in newly formed soils ranges from 0.90% in primary soils of potassium-textured quarries to 6.27% in organo-mineral horizons in Leptosol formed on limestone [33]. In the typical urban soils of Ukhta, its content varies from 2 to 12% with an average value of about 7% [34]. These soils are located on lawns along highways and are characterized by high pH values and contents of alkaline elements. The soils of the city of Arkhangelsk have a carbon concentration of about 2% in the natural soils around the city and 6.8% in the industrial zone of this city [35]. Natural soils have an acidic pH (4.81), while industrial soils are characterized by a near-neutral reaction (pH = 6.37). Another work on the soils of Arkhangelsk [36] notes an increase in carbon content from 0.64% in natural soils to 3.19% in soils constructed by the transport method. The increase in carbon content occurs simultaneously with an increase in the pH level and the content of trace elements. Probably, organic matter accumulates trace elements, and the neutral reaction of pH makes them less mobile in the urban environment. This indicates the regulatory function of organic matter in relation to the main inorganic pollutants.

As for Yakutia (Eastern Siberia), few data were obtained for Tiksi, Chersky, and Yakutsk in a comparison with the Tazovsky and Gaz-Sale settlements in the Yamal region [37]. In the urban soils of Yakutia, the O, T, A, and buried [A] horizons have been documented as the result of urban pedogenesis. There are no detailed data on total organic carbon in this manuscript, but there are numerous interesting data on the accumulation of 15 individual PAHs in urban and industrial soils. Thus, Arctic cities are not clean and pristine in terms of organic pollution, and polycyclic aromatic compounds may play an important role in and contribute to the functioning of the entire soil organic matter system [38]. Values of PAHs in soils are measured in tens of thousands of ng g, but the content of petroleum hydrocabonates can reach tens of mg kg in urban soils, as was noted for the Komi Republic, for which Lodygin [39] prepared a detailed map of the mass concentration of hydrocarbons. For the soils of Anadyr, Magadan, and Yakutsk, the content of resistant organic pollutants was investigated, and it was found that it is not a serious danger; meanwhile, there is an evaporation of these organic substances from the soil surface and movement with soil particles into the air masses, which confirms the requirement of soil quality standardization, including on the migration-air harmfulness indicator [40]. The introduction of spruce to the city of Magadan in the Soviet period was not successful due to unfavorable pedological soil conditions, although in the suburbs of Magadan, it took root in the place of former larch forests [41]. One of the reasons for this phenomenon may be the soil temperature regime, which is more favorable in forests with pronounced snow cover than in Eastern Siberian cities [42]. This may also affect the level of organic matter stabilization, which, however, has not yet been investigated for Arctic cities.

Another approach in characterizing soil organic matter functions and kinetics is in the manuscript of Bobrik et al. [43], which resumed the emission rates of carbon dioxide and carbon of microbial biomass and provides the spatial patterns of their distribution. Soil respiration in abandoned urban and industrial ecosystems of Vorkuta, Naryan-Mar, and Spitsbergen was investigated under the influence of anthropogenic factors [44]. In this work, it was shown that anthropogenic factors can lead to both the intensification of greenhouse gas emissions and their reduction. Thus, the response of cryogenic soils to

anthropogenic input may be highly variable and should be investigated by a wider group of samples representing a wide range of geographic conditions.

Huge concentrations of slag and anthropogenic material in dumps and tailings—up to 91%—have been reported for the city of Norilsk [45], which may play an important role in the increase in carbon content in urban soils due to the movement of anthropogenic carbon by wind and water erosion. This paper highlights that soil respiration is higher in more technogenically disturbed areas; so, organic matter mineralization varies spatially and depends on soil nutrient enrichment in an urbanized ecosystem. The accumulation of oil and petroleum products in soils can cardinally change the pattern of vertical distribution of organic matter, the degree of soil hydrophobicity, and the C/N ratio in soils of the European Arctic [46].

A detailed study of the percentage of carbon and nitrogen in the soil and their profile distribution was published by Nizamutdinov et al. [47] for the central part of the Yamal region (Salekhard agglomeration). It was shown that the morphology of the upper soil horizons in urban soils can vary greatly depending on the type of urban soil: Podzols, Entic Podzols, urban Technosols, and Cryosols. Carbon content was very high in garden soils from 0 to 23%, while the average was about 3–7%. The C/N ratio was more or less comparable for urban and mature soils and ranged from 7 to 26 units. Thus, the urban soils of the central part of the Yamal region combine the properties of natural and anthropogenic soils (a mixture of natural and anthropogenically transformed horizons and anthropogenic artifacts). A different situation was noted for Yakutsk [28], where the percentage of carbon and nitrogen content was lower, which may be related to the replacement of the original soil with a new, transported soil, where the processes of soil formation and carbon accumulation start from the zero point. If the soils belong to the suburban areas, there is no serious transformation of the structural composition of organic matter compared to the undisturbed cryogenic tundra soils [48]. Other data were published by Nizamutdinov et al. [22] for the soils of the town of Nadym, whereby a significant transformation of humic substances during urbanization was observed, which was well expressed in an increase in the aromatic part of humic substances by aliphatic substances, supposedly due to mineralization of the aliphatic compounds. Limited information is known about the soils of Magadan, which belongs to the Eastern Siberian permafrost taiga region. The natural and climatic features of the territory determined the local distribution of agricultural activity, which develops mainly near the settlements of the southwestern part of the region [49]. The content of humus in arable soils varies from 6.9 to 9.0% of its mass, which is relatively high and indicates that the 50-year history of soil development has led to a significant accumulation of humus. The reason for the high content of humus in the cultivated horizon is also the low rate of humification of organic matter in cold climates [22].

Thus, the molecular composition of organic matter in Arctic urban soils is still very poorly studied; there are fragmentary works, sometimes confirming each other and sometimes not. This is typical for the Eurasian cryolithozone and leads to significant errors in the estimation of reserves, not to mention the qualitative composition, of soil organic matter [9]. The last type of organic matter is that produced in numerous abandoned greenhouses around cities; fragmentary information about this type of organic accumulation was presented by Nizamutdinov et al. [22].

In single industrial cities located in the permafrost zone, soil formation in post-technogenic landscapes is extremely slow. In the vicinity of Mirny (Yakutia), to accelerate the reclamation of overburden rock dumps, Technosols were formed where an organic-accumulative horizon was developed over 35 years. The decomposition of organic matter is limited by insufficient development of destructor organisms [50]. In the surface layers of these Technosols, the total carbon content is 1.79%, with an average of 0.75% for the profile. The high C/N ratio of 39.5 indicates insufficient mineralization of organic residues [51]. The humus content in undisturbed zonal soils can reach 7–8%.

The data on the types of sources of soil organic matter are summarized in Table 1. The listed sources of carbon organic compounds are variable in their origin and functional

significance for the formation of the organic matter system of urban soils, but they are all important for understanding the polygenetic and polychemical nature of humus. It is obvious that Arctic urban soils are closest to the real model of future transformations of the permafrost humosphere under the impact of urbanization and industrialization of the tundra and forest tundra. Therefore, the system of methods for monitoring the humus state should include not only the measurement of humus content but also soil density, as well as the origin of organic matter and its recent history.

**Table 1.** Types and functions of organic matter in urban soils.

| The Type of Organic Matter in Urban Soils | Features of the Types of Organic Matter |
| --- | --- |
| Humus inherited from previous soil | This type of organic matter can be part of both active and passive carbon pools, can be relatively quickly mineralized, and can be for a long time preserved in the environment. |
| Newly formed colloidal humus | One of the most stable types of organic carbon in the ecosystem, it is part of the passive carbon pool. |
| Folic moor and moder materials | The carbon of plant residues in various degrees of transformation; part of the active carbon pool. |
| Organic matter migrating vertically due to cryogenic mass transfer | During cryoturbation, it accumulates at the permafrost boundary and is stored there, but as a result of climate change, it can move from a passive pool to an active one and have a significant impact on carbon dioxide emissions from the soil. |
| Manure and peat, used for soil construction and greening | Part of the active carbon pool and the main source of nutrients in urban soil |
| Oil hydrocarbons | These are capable of persisting in the soil for a long time, are difficult to remove, disturb the ecological functions of soils, and negatively affect human health. |
| Persistent organic pollutants | These are capable of persisting in the soil for a long time, are difficult to remove, disturb the ecological functions of soils, and negatively affect human health. |
| Polycyclic aromatic hydrocarbons | These are capable of persisting in the soil for a long time, are difficult to remove, disturb the ecological functions of soils, and negatively affect human health. |

### 3.3. Stocks of Organic Matter Is Urban Soils

Soil organic matter stocks (bulk concentrations) are reported less frequently than gravimetric concentrations (mass percentages). Thus, a paper [52] reports a high variability of stocks from 5 to 27 kg/m$^2$ for background soils in Yamal within a 1 m deep soil profile. This is within the estimate provided by Mishra et al. [2] for a broader range of sample sites in the cryolithozone. The ability to calculate weight concentrations of carbon into volume concentrations is limited by the lack of information on soil density [9]. This is the main lack of soil studies conducted in Siberia. Thus, we lose the opportunity to assess the most important ecological function of soils—the depositing capacity in relation to nutrients—and become limited in the possibilities of accurate assessment and subsequent monetization of the carbon ecosystem services of the cryobiosphere. One way to address this problem is to make volumetric soil density sampling mandatory along with gravimetric soil density sampling. These recommendations are unlikely to be implemented. Another way is to use pedotransfer functions that allow the calculation of soil density from various soil parameters, but even for natural conditions with a large number of field and laboratory studies tested, pedotransfer functions do not show high or even medium efficiency [53–55]. Therefore, this paper can only conclude that there is a strong variability of organic matter stocks in the soils of Arctic cities, which is determined by the diversity of organic carbon input, accumulation, and stabilization in soils and urbanized ecosystems in general.

### 3.4. Climate Change and Carbon Sequestration in the Urbanized Arctic

The number of articles aimed at studying the processes of climate change in the Arctic, as well as carbon sequestration by Arctic biomes, increases every year. About 80% of all publications on carbon topics in the Arctic have been written in the last 11 years [56]. This

indicates the importance of research and the relevance of soil organic carbon stocks and quality in the circumpolar belt [57]. However, most of the research has been conducted in the natural environment, while the urban environment is essentially unexplored [58]. In the natural environment of the Arctic, the main carbon sinks are peatlands, sequestration that occurred through the growth of sedges, mosses, shrubs, and trees in the forest-tundra and taiga zones [59]. As a result of the city's development in the Arctic in the Soviet period, there was a radical change in landscapes in the locations of future cities; peat bogs were drained and the natural plant communities were replaced by more southern species [60]. Thus, anthropogenic influences shifted some natural ecosystems from pure carbon sinks to carbon-neutral or atmospheric carbon sources [61]. The Arctic region is thought to be a net carbon sink, accumulating about 0.13 Pg C-$CO_2$ year$^{-1}$ annually [62]. However, these data are typical for natural systems, which have a localized character, and do not take into account anthropogenically transformed systems. For example, Alaska is a net source of carbon and on average emits 0.025 Pg C-$CO_2$ year$^{-1}$ [63]; for the urbanized areas of Siberia, no such studies have been conducted, so it is quite difficult to consider the contribution of the urbanized systems of the Russian Arctic to global climate change. In the Russian Arctic, there are three major cities located above the Arctic Circle: Murmansk (with an area of 154.4 km$^2$), Norilsk (with an area of 23.2 km$^2$), and Vorkuta (with an area of 29.7 km$^2$); these cities are the largest industrial centers of Russia and sources of greenhouse gases [64]. The area of the largest cities and adjacent territories in the Arctic region of Russia is about 671.96 thousand km$^2$ [65]. The area of the cryolithozone in the world is 35 million km$^2$, and in Russia, the Arctic region occupies 4.7 million km$^2$. Thus, in the Arctic region of Russia, up to 14.26% of the territory is most vulnerable to climate change, as well as carbon hotspots in permafrost. Carbon stocks in the meter-deep layer of soil in the Arctic and Subarctic of Russia amount to about 105 Pg C [66]; thus, in the urbanized territories of the Arctic region of Russia, on average, about 14.97 Pt C is stored at a depth of up to 1 m. The quality of this carbon storage and the rate of its transformation are almost unexplored.

According to predictions, within a century, net carbon emissions could range from 37 to 149 Pg C-$CO_2$ year$^{-1}$ in the cryolithozone under different scenarios, which would negatively affect both natural and urbanized areas [56]. Urban environments are more vulnerable to climate change as a result of vegetation disturbance, permafrost degradation, and landscape transformation. As a result of warming, thermokarst processes caused by the thawing of permafrost, disturbance of the hydrological regime will actively develop, which may lead to drying of the Arctic soil [67]. According to some data [68], the predicted "methane bomb" may not happen because the qualitative composition of soil organic matter varies quite widely in the Arctic and greenhouse gas emissions, as well as the transformation of organic matter, may proceed at a quite slow rate and make insignificant changes to the atmosphere [56]. If the soil cover dries as a result of permafrost degradation and a decrease in the upper border of the water table, there may be a suppression of soil microbes and, accordingly, a decrease in greenhouse gas emissions from the soil [69]. The urbanized areas of the Arctic are an important component of global climate change and are likely to be the most damaged.

## 4. Conclusions

During the review of the various published data and the results of our own studies, the carbon concentration percentages of organic compounds in the soils of urbanized ecosystems of the Russian Arctic were established, which was done for the first time. An analysis of organic matter stocks in the soils of urbanized ecosystems and trends in the profile distribution of carbon in the soils of the cryolithozone was also carried out. As a result of this study, the following conclusions were made:

(1)    It has been revealed that the content and stocks of organic matter in the soils of urbanized ecosystems of the Russian Arctic are poorly studied, even less than the organic matter of the soils of the background (mature) tundra and forest-tundra ecosystems. The carbon content is highly variable and depends mainly on anthropogenic factors

(the accumulation of "industrial" carbon and pyrogenic compounds), as well as on the age of soil formation and soil parent material types.

(2) Organic matter accumulation horizons of urban soils differ in morphology and composition from background soils. Thus, in urban soils, forest litter disappears, and a humus horizon appears, sometimes an arable organo-mineral horizon; if there is a peat (Histic) horizon in the soil, it becomes more drained and mineralized. There are also Technogenic soil-like horizons, consisting mainly of organic matter of artificial origin.

(3) In the composition of organic matter of urban soils, small in mass but important in biogeochemical functions are components such as petroleum hydrocarbons and polycyclic aromatic hydrocarbons. The former is the most studied for a variety of ecosystems of the Komi and the latter for the soils of the Yamalo-Nenets Autonomous Okrug.

(4) The stock of organic matter in the upper horizons of urban soils in the Arctic is poorly connected functionally with the carbon accumulated in the deeper layers of native soils; due to the young age of the cities, the cryogenic mass exchange has not yet led to the connection of these pools.

(5) Urbanized areas of the Arctic are practically unexplored in terms of carbon emission and sequestration potential. It is generally accepted that the Arctic is a net sink of carbon and stores about 0.13 Pg C-$CO_2$ year$^{-1}$ annually, but the example of Alaska revealed that this area is a net source of carbon in the Arctic, 0.025 Pg C-$CO_2$ year$^{-1}$. No such studies have been conducted in Siberia, which leads to a serious underestimation of these territories and their contribution to global climate change; therefore, a study of the carbon pool and morphological features will make it possible to predict future scenarios of climate change and greenhouse gas emissions and calculate the sequestration potential of the Arctic territories.

**Author Contributions:** Conceptualization, E.A., V.P. and A.P.; resources, A.P.; data curation, E.A., V.P., A.P. and T.N.; writing—original draft preparation, E.A., V.P., A.P. and T.N.; writing—review and editing, E.A., V.P. and A.P.; visualization, E.A. and A.P.; supervision, E.A. All authors have read and agreed to the published version of the manuscript.

**Funding:** This research was funded by Saint Petersburg State University, project ID: 101662710 (CZ_MDF-2023-1).

**Institutional Review Board Statement:** Not applicable.

**Informed Consent Statement:** Not applicable.

**Data Availability Statement:** Not applicable.

**Acknowledgments:** The article is dedicated to the 300th anniversary of the founding of St. Petersburg State University.

**Conflicts of Interest:** The authors declare no conflict of interest.

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
