# Peer review of "Soil Organic Matter in Urban Areas of the Russian Arctic: A Review"

_atmosphere, doi:10.3390/atmos14060997_

Round 1

Reviewer 1 Report

Atmospheres

Title: The role of soil organic matter in green house gazes state of urbanized ecosystems of the Russian Arctic. A review

General Comments

The paper is a literature review on soil carbon in urbanized areas of the Russian Arctic. The review is interesting. 

The paper is readable, and I only have editorial comments. 

Specific Comments

1)    The Title needs an edit: ‘Soil organic matter in urban areas of the Russian Arctic: A review.’ Consider deleting ‘greenhouse gases’ from the title since it is not a major focus of the review. You could use ‘soil carbon stocks’ rather then ‘soil organic matter.’

2)    I suppose urban areas are ‘ecosystems’, per se, but ‘urban areas’ is a better term. 

3)    The Abstract is mostly okay. However, try to articulate a better conclusion in the last sentence. Rather than say ‘more work is needed’ try to articulate a specific recommendation. 

4)    I do not agree your indication that dichromate oxidation is an ‘indirect’ method. Before the advent of commercial combustion analyzers, dichromate oxidation was the standard way to quantify soil organic matter. 

5)    In the Introduction, describe ‘K’ in more detail; ‘empirical coefficient’ is not specific enough. Indeed, I am not sure what it means. 

6)    The Methods section needs more detail. Describe the soil measurements you made. Also include how you searched the literature. You need to give enough information for a reader to repeat the study. For example, describe micromorphology in the Methods. 

7)    A map of Russian showing the location of the cities would be helpful. 

8)    Also, it would help readers if you also include references to the World Reference Base (WRB), FAO system for soil classification (https://www.fao.org/soils-portal/data-hub/soil-classification/world-reference-base/en/). Not all readers will know the Russian system. 

9)    Is all the black carbon from coal burning? Do fires add black carbon. Add a bit more detail in the origin of black carbon.

Technical Comments

1)    Line 32: change ‘everywhere from the mountaintops to ocean depths’ to ‘globally.’

2)    Line 33 – 35: maybe delete. Just say, ‘climate is changing, and permafrost soils are crucial to future climate change.’

3)    Line 40: start a new paragraph. Add a topic sentence, such as ‘although permafrost soil store carbon, the amount is variable. For example, etc.’

4)    Line 53 and elsewhere: delete ‘at the same time’ or say ‘currently.’

5)    Line 102: add ‘and cold temperatures.’

6)    Line 124: new paragraph.

7)    Line 188: change ‘taken into account’ to ‘noted.’

8)    Line 189: change ‘it should be noted’ to ‘notably.’

9)    Line 209: start a new paragraph, or on line 214.

10) Line 231: start a new paragraph. 

11) Line 256: describe how you made the distinction between ‘natural’ and ‘newly.’

12) Figure 3: convert to a Table.

In a few cases, word choice is a bit awkward. 

Author Response

Response to a review of the manuscript “The role of soil organic matter in green house gazes state of urbanized ecosystems of the Russian Arctic. A review.

Dear reviewer!

Thank you for your comments, they were completely taken into account, which improved the quality of the article for publication in Journal.

Text that has been changed is marked by yellow color.

General comments:

  1. The Title needs an edit: ‘Soil organic matter in urban areas of the Russian Arctic: A review.’ Consider deleting ‘greenhouse gases’ from the title since it is not a major focus of the review. You could use ‘soil carbon stocks’ rather then ‘soil organic matter.’

Response: Thank you! We have changed the title.

  1. I suppose urban areas are ‘ecosystems’, per se, but ‘urban areas’ is a better term.

Response: Thank you! It has been changed.

  1. The Abstract is mostly okay. However, try to articulate a better conclusion in the last sentence. Rather than say ‘more work is needed’ try to articulate a specific recommendation.

Response: Thank you! The abstract has been reworked.

  1. I do not agree your indication that dichromate oxidation is an ‘indirect’ method. Before the advent of commercial combustion analyzers, dichromate oxidation was the standard way to quantify soil organic matter.

Response: Thank you! It has been changed.

  1. In the Introduction, describe ‘K’ in more detail; ‘empirical coefficient’ is not specific enough. Indeed, I am not sure what it means.

Response: Thank you! We added additional information in Introduction section.

  1. The Methods section needs more detail. Describe the soil measurements you made. Also include how you searched the literature. You need to give enough information for a reader to repeat the study. For example, describe micromorphology in the Methods.

Response: Thank you! We added additional information.

  1. A map of Russian showing the location of the cities would be helpful.

Response: The map has been added.

  1. Also, it would help readers if you also include references to the World Reference Base (WRB), FAO system for soil classification (https://www.fao.org/soils-portal/data-hub/soil-classification/world-reference-base/en/). Not all readers will know the Russian system.

Response: Thank you! We added the name of soils in WRB classification.

  1. Is all the black carbon from coal burning? Do fires add black carbon. Add a bit more detail in the origin of black carbon.

Response: We added additional information.

Minor comments:

  1. Line 32: change ‘everywhere from the mountaintops to ocean depths’ to ‘globally.’

Response: done

  1. Line 33 – 35: maybe delete. Just say, ‘climate is changing, and permafrost soils are crucial to future climate change.’

Response: done

  1. Line 40: start a new paragraph. Add a topic sentence, such as ‘although permafrost soil store carbon, the amount is variable. For example, etc.’

Response: done

  1. Line 53 and elsewhere: delete ‘at the same time’ or say ‘currently.’

Response: done

  1. Line 102: add ‘and cold temperatures.’

Response: done

  1. Line 124: new paragraph.

Response: done

  1. Line 188: change ‘taken into account’ to ‘noted.’

Response: done

  1. Line 189: change ‘it should be noted’ to ‘notably.’

Response: done

  1. Line 209: start a new paragraph, or on line 214.

Response: done

  1. Line 231: start a new paragraph.

Response: done

  1. Line 256: describe how you made the distinction between ‘natural’ and ‘newly.’\

Response: done

  1. Figure 3: convert to a Table.

Response: done.

Thank you for work of our article.

Sincerely,

Professor of Saint-Petersburg State University, Evgeny V. Abakumov.

Reviewer 2 Report

The reviewer is really grateful to the authors for such an outstanding, modern and vital research and article about the role of soil organic matter in green house gazes state of urbanized ecosystems of the Russian Arctic. Since polar ecosystems are the most important storage and source of climatically active gases and at the same time, natural biogeochemical processes of organic matter circulation in the soil-atmosphere system are disturbed in urban ecosystems of the cryolithozone so urbanized eco-systems in the Arctic are extremely under investigated in terms of their functions in regulating the cycle of climatically active gases. The percentage or gravimetric concentrations of organic matter in Arctic urban soils are almost always determined by the in-direct method of dichromate oxidation and are subject to extreme variability (from tenths of a percent to more than 90 % in man-made soil formations), but the average carbon content in the surface soil horizons can be estimated at 5-7 %. The surface humus-accumulative horizons are represented by a variety of morphological forms with the content of organic matter of various origin. The work also focuses on those forms of organic matter, the content of which is extremely small, but very important for the biogeochemical functioning of soils - polycyclic aromatic hydrocarbons and components of petroleum products, as well as labile forms of soil organic matter. It is stated that further work on the organic matter system of urban soils in the Arctic is needed, because the carbon cycle here is severely disrupted, carbon flows are severely disrupted, and urbanization and industrialization in the Arctic are progressing, which could lead to a radical transformation of carbon ecosystem services.

So the main question addressed by the research is the role of soil organic matter in green house gazes state of urbanized ecosystems of the Russian Arctic. The paper is a review. The considered topic is original, relevant and very significant in the field especially in the frame of the modern climatic change. The conclusion is consistent with the evidence and arguments presented and do address the main question posed. The references are appropriate.

Author Response

Response to a review of the manuscript “The role of soil organic matter in green house gazes state of urbanized ecosystems of the Russian Arctic. A review.

Dear reviewer!

Thank you for your review, we make some changes in the article which is improve the quality. It’s marked by yellow color.

Thank you for work of our article.

Sincerely,

Professor of Saint-Petersburg State University, Evgeny V. Abakumov.

Reviewer 3 Report

Article discussed the influence of organic matter of green house gases, on urbanized ecosystems of the Russian Arctic. It is proved that ecosystems in the Arctic are extremely  under investigated, concerning regulating the cycle of climatically active gases. More work should be provided to avoid progressing of industrialization in the Arctic, what could lead to radical transformation of carbon ecosystem services.

Please provide separate Section - Discussion.

Section Conclusions please devide on Subpoints, to present the most important findings of the study.  

Minor editing of English language required.

Author Response

Response to a review of the manuscript “The role of soil organic matter in green house gazes state of urbanized ecosystems of the Russian Arctic. A review.

Dear reviewer!

Thank you for your comments, they were completely taken into account, which improved the quality of the article for publication in Journal.

Text that has been changed is marked by yellow color.

General comments:

  1. Please provide separate Section - Discussion.

Response: Thank you! We would like to keep the Results and Discussion section, since the article is a Review in which we have combined both our own results and a review of the various literature sources. We have worked to correct a number of comments in the article, we hope that after that it will improve its quality.

  1. Section Conclusions please divide on Subpoints, to present the most important findings of the study.

Response: Thank you! It’s done.

Thank you for work of our article.

Sincerely,

Professor of Saint-Petersburg State University, Evgeny V. Abakumov.

Reviewer 4 Report

this manuscript make a review along with a survey between 2013-2022, aimin to analysis the organic carbon content  status in russian arctic cities. i think its well written ,however its more like a report instead of research paper or review. 

anyhow ,its ok to publish,  

comments:

1.topic: green house gazes?  or greenhouse gases?

2. I suggest , a map with locations of  different sampling sites was needed .  because most of the readers dont know much about the research area.

3. figure 3 is useless, you just need to list the sources of organic matter in the text of the manuscript .

Author Response

Response to a review of the manuscript “The role of soil organic matter in green house gazes state of urbanized ecosystems of the Russian Arctic. A review..”.

Dear reviewer!

Thank you for your comments, they were completely taken into account, which improved the quality of the article for publication in Journal.

Text that has been changed is marked by yellow color.

General comments:

  1. .topic: green house gazes? or greenhouse gases?

Response: The title has been changed.

  1. I suggest , a map with locations of different sampling sites was needed .  because most of the readers dont know much about the research area

Response: We added the map of sampling sites.

  1. figure 3 is useless, you just need to list the sources of organic matter in the text of the manuscript .

Response: Thank you! We replaced Figure 3 with a Table 1 describing the types and functions of organic matter in the urban area.

Thank you for work of our article.

Sincerely,

Professor of Saint-Petersburg State University, Evgeny V. Abakumov.